# A Potential Biomarker of Brain Activity in Autism Spectrum Disorders: A Pilot fNIRS Study in Female Preschoolers

**DOI:** 10.3390/brainsci13060951

**Published:** 2023-06-14

**Authors:** Elena Scaffei, Raffaele Mazziotti, Eugenia Conti, Valeria Costanzo, Sara Calderoni, Andrea Stoccoro, Claudia Carmassi, Raffaella Tancredi, Laura Baroncelli, Roberta Battini

**Affiliations:** 1Department of Neuroscience, Psychology, Drug Research and Child Health NEUROFARBA, University of Florence, 50135 Florence, Italy; 2IRCCS Stella Maris Foundation, Viale del Tirreno, 56128 Pisa, Italy; 3Institute of Neuroscience, National Research Council, Via Moruzzi 1, 56124 Pisa, Italy; 4Department of Clinical and Experimental Medicine, University of Pisa, Via Roma 67, 56126 Pisa, Italy; 5Department of Translational Research and of New Surgical and Medical Technologies, University of Pisa, 56100 Pisa, Italy

**Keywords:** autism, female, preschoolers, fNIRS, biomarker, visual physiology

## Abstract

Autism spectrum disorder (ASD) refers to a neurodevelopmental condition whose detection still remains challenging in young females due to the heterogeneity of the behavioral phenotype and the capacity of camouflage. The availability of quantitative biomarkers to assess brain function may support in the assessment of ASD. Functional Near-infrared Spectroscopy (fNIRS) is a non-invasive and flexible tool that quantifies cortical hemodynamic responses (HDR) that can be easily employed to describe brain activity. Since the study of the visual phenotype is a paradigmatic model to evaluate cerebral processing in many neurodevelopmental conditions, we hypothesized that visually-evoked HDR (vHDR) might represent a potential biomarker in ASD females. We performed a case-control study comparing vHDR in a cohort of high-functioning preschooler females with ASD (fASD) and sex/age matched peers. We demonstrated the feasibility of visual fNIRS measurements in fASD, and the possibility to discriminate between fASD and typical subjects using different signal features, such as the amplitude and lateralization of vHDR. Moreover, the level of response lateralization was correlated to the severity of autistic traits. These results corroborate the cruciality of sensory symptoms in ASD, paving the way for the validation of the fNIRS analytical tool for diagnosis and treatment outcome monitoring in the ASD population.

## 1. Introduction

Autism spectrum disorder (ASD) is a complex neurodevelopmental condition characterized by early-onset and persistent challenges in social interactions, impaired communication, restricted interests, repetitive behavior, and alterations of sensory processing [1]. Despite substantially heterogeneous data across studies, recent evidence highlights that the average worldwide prevalence of ASD is one in 100 children [2], indicating that this disorder is a major burden for health-care systems and society. Male predominance is a consistent finding in ASD, with a male/female ratio of 4–5:1 [3]. Despite the lack of an effective cure for ASD, several therapeutic interventions have been shown to improve the child’s functioning [4,5]. In this framework, a precocious and accurate diagnosis is a vital step in maximizing the beneficial effect of tailored interventions on clinical outcomes [6]. However, the early prediction of the ASD phenotype is still challenging for clinicians due to the multidimensional nature of this condition, the heterogeneity of symptom severity, their developmental changes across the lifespan, and the variability between male and female presentation [7,8,9]. Since the evaluation of developmental history and the phenotypic observation of behavioral readouts can be highly prone to subjective bias, the search for reliable, objective and quantitative biomarkers to assess the function of cerebral circuits represents an important advancement to refine the diagnostic and therapeutic pipeline in ASD [10]. This is particularly true for females that are usually less likely to receive a timely ASD diagnosis than males, due to diverse phenotypes, psychiatric co-morbidities, and high capacities for camouflage, even at comparable levels of autistic symptoms [11,12,13,14,15]. In addition, this cohort of ASD patients has been sorely under-investigated in the scientific literature [16].

In recent years, the biological dimension of ASD has been extensively investigated, using advanced tools to study the morphology and dynamic processing of brain circuits, such as high-density EEG, Magnetic Resonance Imaging (MRI), and Magnetoencephalography (MEG) [17,18,19,20]. Atypical neuroanatomical and neurofunctional features have been largely reported in the brains of ASD subjects, using imaging and electrophysiological methods [21,22,23,24,25,26], suggesting that an ASD-related signature might be recognized in the neural circuits. To date, the high complexity of data collection and analytic algorithms, however, significantly hampers the incorporation of these instrumental readouts in the routine diagnostic pathway.

Among emergent methods, functional near-infrared spectroscopy (fNIRS) is a non-invasive and undemanding technique that provides an indirect measure of neural activity through the quantification of changes in the concentration of different species of hemoglobin [27]. The high portability and experimental flexibility make fNIRS a powerful imaging tool for very young children, bringing functional studies of brain circuits into a more naturalistic environment [28,29]. Resting-state and task-evoked fNIRS recordings have been previously used to evaluate multiple aspects of brain connectivity and function in the ASD population [30,31]. Even in preschoolers, a considerable amount of research groups employed fNIRS to investigate brain activation patterns underlying the core symptoms of ASD (for a recent review, see Conti et al., 2022 [32]). These studies showed the hypoactivation of frontotemporal regions in response to social perception tasks [33,34], and the atypical processing of the temporal area during language-related tasks [35] in ASD children versus typically developing (TD) peers. Moreover, an alteration of global and local functional connectivity has been reported in preschoolers with ASD, suggesting an inefficient transfer of information among cortical regions [36,37].

More broadly, there is a growing interest in the application of fNIRS to define sensitive biomarkers of brain function in order to support the clinical assessment and monitor the developmental trajectories in neurodevelopmental disorders, including ASD [29,31,32]. In this context, sensory symptoms of ASD can be detected early in childhood, possibly predicting later diagnostic status [38,39], and the study of the visual phenotype is a paradigmatic model to evaluate the cerebral processing in many other developmental conditions [40,41,42,43,44]. Thus, we hypothesized that visually-evoked hemodynamic responses (vHDR) might represent a quantitative and unbiased biomarker to assist the clinical assessment of ASD, in particular for the challenging female phenotype. 

We performed a pilot study on a cohort of high-functioning preschooler females with idiopathic ASD (fASD) with two specific aims: (i) to test the feasibility and reliability of a novel standardized procedure of visual stimulation with a high entertaining value [45] to measure vHDR in this population; and (ii) to investigate the validity of vHDR as an effective tool to discriminate between preschool girls with TD and sex/age matched peers with ASD.

## 2. Materials and Methods

### 2.1. Subjects

We performed an observational prospective monocentric study in a tertiary care university hospital (the IRCCS Stella Maris Foundation, Pisa, Italy). We recruited 12 females with idiopathic ASD (fASD: median age 4.5 years; SD: 1.17) and 13 sex/age-matched control typically developed peers (TD: median age 4.86; SD: 1.17). Established inclusion criteria were: (1) a diagnosis of ASD according to the DSM-5 criteria (APA, 2013); (2) age-range: between 3 and 6 years; (3) female gender; (4) a negative history for premature birth or neurologic complications possibly related to delivery; and (5) non-verbal IQ > 70. An ASD diagnosis was performed by a multidisciplinary team including a senior child psychiatrist, an experienced clinically trained research child psychologist, and a speech language pathologist over 5–7 days of extensive evaluation. The demographic characteristics and clinical features of experimental cohorts are listed in Table 1 and Table 2. Clinical measures such as the Autism Diagnostic Observation Schedule (ADOS) [46], the children’s version of the Autism Questionnaire (AQ) [47], and a cognitive and adaptive profile were systematically collected for each subject using internationally validated scales/interviews, such as the Wechsler Preschool and Primary Scale of Intelligence [48] and the Vineland Adaptive Behavior Scales–II–VABS [49]. 

All participants reported normal or corrected-to-normal vision. The experimental procedures were authorized by the Regional Pediatrics Ethics Board (Comitato Etico Pediatrico Regionale-Azienda Ospedaliero-Universitaria Meyer-Firenze, Italy; authorization number 119/2021 (13 April 2021), and were performed according to the declaration of Helsinki. Written informed consent was obtained from the parents of each child, authorizing the use of anonymized data for research purposes. 

### 2.2. Apparatus and Experimental Design

We employed a NIRSport system (8 × 8, NIRx Medical Technologies LLC, Berlin, Germany) consisting of eight red light-sources operating at 760 nm and 850 nm, and seven detectors placed into a textile EEG cap, forming an array of 22 channels centered on occipital cortical areas, as previously described [45]. Different sizes of the cap were used according to the head circumference of the subject. For data recording, Aurora Software 1.4.1.1 (NIRx Medical Technologies LLC) was used. The sampling rate was 10.2 Hz. We measured hemodynamic signals in response to a radial checkerboard-blended cartoon freely chosen by the subject [45]. Each experimental session lasted for approximately 20 min, including the calibration of light coupling between sensors and detectors. Visual stimuli were generated using Python 3, Psychopy3, and open CV [50,51]. Visual cortical hemodynamics in response to full-field, reversing, square wave, and radial checkerboard (RC), with abrupt phase inversion (0.33 c/deg, 4 Hz), were evaluated in the time domain by measuring the peak-to-baseline amplitude. We used an event-related design consisting of: (i) 20 trials of 5 s stimulus ‘on’ followed by 10 s stimulus ‘off’ and (ii) 20 cycles of 5 s mock stimulus ‘on’ followed by 10 s stimulus ‘off’. The stimulus ‘on’ was a merge between the RC (90% of contrast) and an animated movie, whereas the stimulus ‘off’ was the grey-scale isoluminant baseline video. The mock stimulus ‘on’ consisted of the merging of the RC at 0% of contrast. The baseline cartoon was presented at 80% of contrast. The two stimulating conditions were pseudo randomly interleaved for each subject during the recording. Each experimental block lasted 10 min (a total of 40 trials). Visual events were synchronized with NIRSport over wireless LAN communication through the Python version of the LabStreamingLayer (https://github.com/sccn/labstreaminglayer, accessed on 11 August 2022. The experimental protocol did not include gaze tracking during visual stimulation. In order to mitigate this critical aspect, a member of the clinical staff assisted the children during the full length of the recordings. No significant distractions from the video were reported. At the end of each experimental session, data were quickly analyzed and visualized using nirsLAB software (NIRx Medical Technologies LLC., v2019.4) in order to assess the data quality.

### 2.3. Signal Processing

Data preprocessing was completed using the Homer3 package (v1.29.8) in MATLAB (R2020a) and the processing stream was tailored for the detection and correction of motion artefacts, as detailed in Mazziotti et al. [45]. The resulting txt file was imported in Python as a Pandas DataFrame. For each subject, both the channel with the highest response amplitude and the average response across channels were analyzed. The peak was identified as the maximal value for total hemoglobin (THb) and oxygenated hemoglobin (OHb), and the minimum value for deoxygenated hemoglobin (DHb). A grand average was taken of the 20 trials of data per stimulating condition, and differences between visual stimulation ‘on’ (reversing checkerboard) and ‘off’ (blank) were compared. A laterality index was calculated as LI = (peak_left − peak_right)/(peak_left + peak_right), where peak_left was the average maximum value of OHb for the channels in the left hemisphere, and peak_right was the corresponding value for the channels in the right hemisphere (see Conti et al. [52])

### 2.4. Statistical Analysis

A statistical analysis was carried out using Python, as described in Mazziotti et al. [45]. A *t*-test was used to assess (i) the reliability of vHDR in typical children and fASD subjects, (ii) differences in the signal amplitude between the two populations; and (iii) differences in the laterality index. A two-way ANOVA for repeated measures was employed to test differences in OHb peak responses between the two visual cortices of the two populations. We tested the interaction between the signal amplitude of the two hemispheres with a Spearman correlation. The same statistical test was applied to assess the correlation between the amplitude of fNIRS measures and the cognitive/behavioral scores. Adjustments for multiple comparisons were performed using the Benjamini/Hochberg false discovery rate (BH-FDR) correction. All of the plots have been generated using the Matplotlib Python library [53]. All statistical metrics and details are reported in Appendix A.

## 3. Results

### 3.1. A Cartoon-Based Stimulus Is Able to Evoke Reliable Responses in the Visual Cortex of fASD Children

We measured the cortical vHDR elicited by a reversing checkerboard pattern merged with an isoluminant commercial cartoon. In agreement with previous results [45], we obtained a significant activation of the occipital cortex in response to the stimulus in TD children. Grand averages across control participants for Total Hb (THb), Oxygenated Hb, (OHb) and Deoxygenated Hb (DHb) concentration changes are plotted in Figure 1 and Figure 2, showing the average amplitude of signals recorded in the different channels of the montage (Figure 1A–C) and the amplitude of the channel with the highest response within subjects (Figure 2A–C), respectively. A statistical analysis revealed a significant main effect of the checkerboard stimulus (S) with respect to the presentation of the mock stimulus (MS) for all HDR metrics. Importantly, a significant vHDR, with a prominent change of THb, OHb, and DHb concentration in response to the S with respect to the blank for all conditions tested, was also present in the fASD cohort (Figure 1D–F and Figure 2D–F). These results reinforce the validity of the cartoon-based stimulating procedure to evoke a reliable activation of the visual cortex, showing that this method is also suitable to study visual cortical processing in a clinically relevant population, such as fASD children.

### 3.2. The Amplitude of Visual Cortical Responses Is Significantly Lower in the ASD Cohort

We compared the level of cortical activation in response to the stimulus in TD subjects and fASD children. Our data showed that the average amplitude of cortical changes of OHb concentrations was significantly lower in fASD children with respect to the control age-matched cohort (Figure 3B). No significant difference was detected between the two groups for THb and DHb (Figure 3A,C); only a trend towards reduction can be observed for THb of fASD (Figure 3A). Totally comparable data were found in the analysis of the channel with the highest response (Figure 3D–F). In contrast, no change in the signal latency was present between the two groups, except for an increased latency of the DHb peak in fASD (Appendix A).

### 3.3. Atypical Lateralization of Visual Responses in the ASD Cohort

Since one of the main features of autism neurobiology is the lack of lateralization in brain circuits at a structural and functional level [54,55,56], we compared visual cortical responses recorded in the right and left hemispheres for all participants. Our data showed that the average OHb signal detected in the channels mounted upon the right visual cortex of TD children was significantly higher compared to the response recorded through the corresponding optodes of the left hemisphere (Figure 4A,B). Conversely, no inter-hemispheric difference was detected in the cortical activation of fASD children (Figure 4A,B). Notably, the amplitude of signals recorded in the right channels of the TD group was higher with respect to the responses detected in fASD children in the same channels (Figure 4B). Moreover, we calculated a laterality index (LI), i.e., the ratio between the average responses in homologous channels between the two hemispheres. We found that LI is shifted towards the right hemisphere in TD children, thus strengthening the concept of right predominance in the physiological processing of visual stimuli (Figure 4C). Contrastingly, the occipital cortex of fASD children shows a high variability of LI, with some individuals minimally lateralized, some others left-shifted, and a group of right-lateralized as TD controls (Figure 4A–C). Along the same lines, the distribution of vHDR amplitudes recorded in the two hemispheres highlighted a right bias in TD compared to fASD participants (Figure 4D). Finally, a significant correlation between the dimension of OHb signals in the two hemispheres was found in the control group only (Figure 4E).

### 3.4. vHDR Might Be Predictive of Symptom Severity in fASD

Alongside fNIRS, we performed a detailed clinical assessment of fASD children, with a range of parent questionnaires and interviews, in addition to direct evaluation (AQ, VABS, ADOS and non-verbal IQ), in order to provide a robust description of the population studied. Moreover, we tested the correlation between different vHDR metrics and the clinical measures collected. Although no statistically significant effects were detectable, a trend towards a positive correlation between the amplitude of OHb changes and VABS total score was observed (Appendix A). In addition, we found that the level of response lateralization was negatively correlated to the autistic traits of fASD subjects: indeed, the higher the AQ score revealed by the questionnaire, the lower the extent of the right bias in vHDR (Figure 5A). Despite the low numerosity of our sample, we also observed a trend suggesting a possible correlation between LI and the VABS score, indicating that this vHDR measure might also predict the impairment of adaptive functioning (Figure 5B). No correlation was found with ADOS scores and non-verbal IQ (Figure 5C,D).

## 4. Discussion

We performed a cross-sectional observational study, measuring visually-evoked hemodynamic responses (vHDR), in a cohort of high-functioning preschooler females with idiopathic ASD (fASD), and gender-, age-matched typically developing (TD) controls. To this purpose, we used the cartoon-based procedure for visual stimulation that we previously validated in the adult and young general population [45]. The first major finding of the present study is that in all participants, including fASD children, the patterned stimulus was reliable in eliciting a significant change of cortical Hb with respect to the reference baseline, while no response was detected following the presentation of a mock, grey stimulus. As expected from previous studies of evoked vHDR [45,57], we observed a rise in total (THb) and oxygenated Hb (OHb), with a parallel negative change in the concentration of reduced Hb (DHb). Interestingly, the same results emerged from the analysis of the average amplitude of Hb concentration across the 22 channels of our occipital montage, and from the evaluation of the Hb peaks measured in the channel with the highest vHDR within each recording (best channel). These data demonstrate the feasibility of this novel fNIRS procedure in an ASD population with a sub-optimal compliance to structured experimental environments, further establishing the high entertaining value of the cartoon-based stimulation [45] and the ecological merits of the fNIRS technique [28,58]. This observation is totally consistent with previous evidence suggesting the applicability of fNIRS in the study of atypical brain development [29].

The magnitude of vHDR modulation, and in particular of OHb, was significantly different between the two groups studied, with a lower amplitude of OHb response in the visual cortex of fASD subjects compared to TD controls. The selective change of OHb is not surprising considering that this metric is the most commonly used for investigating evoked HDR in the ASD brain [32,59,60]. Moreover, our findings are also consistent with previous studies showing that the activation of cortical circuits negatively reflects autistic traits in TD children [45], but also ASD symptom severity in the clinical population [38,61,62]. Since fNIRS recordings provide a quantitative and reliable readout of brain circuit function [28], and early abnormalities of sensory cortices might precede the onset of socio-communicative and cognitive symptoms in ASD [38,63], our data suggest, for the first time, that vHDR might represent a novel non-invasive, analytical tool to support the diagnostic assessment in females, overcoming the subjective bias intrinsically affecting clinical observations and parental interviews. This provides an important contribution to the field, considering that girls and women with ASD are mostly underserved by clinical criteria and diagnostic processes [64], and that this clinical population is strongly underrepresented in autism research [65]. It is worth noting that the analysis of vHDR might also (i) optimize the follow-up of fASD, tracking the developmental trajectory of altered brain circuits over time; and (ii) provide a reliable protocol to longitudinally monitor, in combination with behavioral testing, the efficacy of tailored intervention strategies. Although previous fNIRS studies on preschoolers suggested the validity of this technique to discriminate between ASD subjects and TD, using both resting-state and task-evoked experimental designs [32,37], the innovative value of our approach relies on the elevated accessibility of visually evoked recordings requiring a relatively low commitment of tested subjects and allowing an easy extraction of fundamental metrics. We need to acknowledge that this is a pilot study on a small sample of subjects, but our results set the background for the validation of this novel paradigm in a larger cohort of fASD children. Indeed, further studies are needed to establish the robustness and specificity of this biomarker. Interestingly, the applicability of fNIRS infants and toddlers [66] might open the possibility in the mid-term to use the vHDR biomarker for early ASD detection, especially in the at-risk population (e.g., children with an older siblings diagnosed with ASD).

Reduced vHDR in fASD children might be ascribable to the differences of visual sensory processing repeatedly observed in the ASD brain. Indeed, a peculiarity in the visual behavior in terms of abnormal eye contact and the atypical processing of faces as well as distinctive perceptual styles, has been frequently reported in ASD, and emerged during the first year of life [38,67,68,69]. In particular, the preference for focusing on local details vs. the global stimulus configuration and low-strength motion perception might determine a less effective activation of the cortical circuits. In agreement with our data, neuroimaging evidence indicates a low-processing-level origin of autistic sensory traits with atypical responses in primary sensory cortices across modalities and the increased inter-trial variability of the evoked responses [38], in addition to occipital structural abnormalities [39,70]. Alterations of the neurochemical and functional architecture of neural circuits detected in the ASD visual cortex [38] might also explain the weaker activity in response to our cartoon-based stimulation, although we cannot exclude a possible contribution of top-down attentional modulation of sensory signaling.

Finally, we also described a predominant activation of the right hemisphere in the control group. A rightward asymmetry of typical visual processing is consistent with the previous literature: indeed, a number of studies combining multiple experimental approaches reported a right hemispheric dominance in typical subjects [71,72,73]. In contrast, we did not detect any specific lateralization in the recordings of fASD children. Accordingly, the loss of brain asymmetry has been widely reported in ASD studies, either in language-related circuits [52] or in non-verbal networks, including both higher cognitive/associative domains and primary sensorimotor function [55]. In the context of the visual system, it is also worth noting that binocular rivalry is weaker in autism [74], and this deficit might be due to the reduced GABAergic action in the visual cortex [75], but also to the reduced strength of interhemispheric connections between the left and right visual cortex [39,76]. Despite the fact that previous fNIRS studies focusing on hemispheric asymmetry in ASD individuals reported controversial evidence [30,32] our data reinforce the notion of atypical lateralization in the ASD brain, expanding the assumed topology to other cortical regions [55]. Since occipital activity is strongly modulated by internal brain states and attentional networks [77,78], we might hypothesize that the asymmetry of typical visual dynamics might arise by a leftward bias of attentional systems to the left hemifield [1,73,79], and that the atypical lateralization in the ASD brain reflects at least in part a centralized deficit in domain-general cognitive processes [38]. Due to the high variability of LI, however, more data are needed to explore the functional differences among the clusters detected.

Interestingly, the laterality index (LI) is predictive of the severity of autistic traits measured which used the children’s version of the Autism-Spectrum Quotient (the Italian version of AQ-child). Indeed, we found that the higher the AQ scores of subjects, the weaker the rightward asymmetry of visual responses, suggesting that this measure within an individual is able to capture the level of autistic phenotype in the fASD cohort. In contrast, we failed to detect a significant correlation between LI and the other observational scales evaluating the extent of the behavioral impairment of subjects. This is likely due to the low numerosity of our sample, but also to the relatively high homogeneity of our cohort and related output range of clinical measures. Indeed, our fASD cohort presented a restricted variation range in the ADOS-2 comparison score [80], suggesting a similar inter-subject symptom severity. Moreover, we detected a correlation trend between LI and the VABS score, even if not statistically significant, suggesting that VABS scores were more capable than non-verbal IQ scores in terms of highlighting the functional impairment of fASD children according to previous studies on this topic [81,82].

## 5. Conclusions

The main strength of this work relies on the study of preschoolers with fASD, representing a sorely under investigated population in autism research [65] and a critical diagnostic challenge for clinicians [15,83]. Our data set the initial background for the use of fNIRS visually evoked recordings as a tool to support the diagnosis and the monitoring of treatment responses in the fASD subpopulation. However, some limitations need to be discussed. First, the small size of the sample recruited (partly due to long-lasting COVID-19 restrictions in Italy) hindered the full validation of fNIRS signals as a biomarker of autistic traits. Second, although our fASD cohort was very well characterized, we need to acknowledge that they represent a very selected sample in terms of both gender and developmental quotient, thus preventing too much generalization of our results to the whole ASD population. Further studies in a larger cohort of subjects, including children with a larger range of symptom severity, are needed in order to assess the sensitivity and the specificity of this analytical tool. In particular, it would also be interesting to more precisely define TD subjects in order to disentangle dimensional contributions (e.g., communication skills, adaptive skills, and neuropsychological competences) in HDR responses. Moreover, it would be beneficial to study the amplitude of fNIRS signals in age-matched males. Indeed, ASD females show a distinctive pattern of resting state EEG activity compared to ASD males [84], suggesting that research concerning biomarkers needs to consider the moderation of the biological sex.

## Figures and Tables

**Figure 1 brainsci-13-00951-f001:**
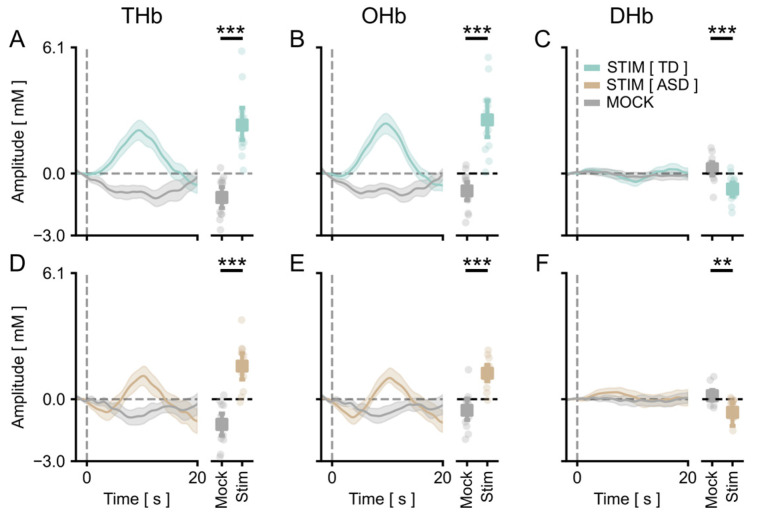
vHDR was reliably detected in TD and fASD participants: average across channels. In all panels, the values in the *y*-axis are multiplied by 10^5^. (**A**–**C**) On the left, the average time course for THb (**A**), OHb (**B**), and DHb (**C**) in response to the radial stimulus (Stim, green line) or the mock stimulus (Mock, grey line) are shown for TD participants. The plots on the right depict the average peak response to Stim vs. Mock across all the subjects. The stimulus-driven signal was significantly higher with respect to the mock condition for all of the metrics (*t*-test, *p* < 0.001 for all comparisons). (**D**–**F**) The same plots as above for the fASD subjects. The stimulus-evoked vHDR was significantly higher with respect to the mock condition (*t*-test, *p* < 0.001 for THb and OHb, *p* < 0.01 for DHb). The orange lines are the Hb responses to the stimulus in fASD plots. The data are expressed as mean ± sem. ** *p* < 0.01, *** *p* < 0.001.

**Figure 2 brainsci-13-00951-f002:**
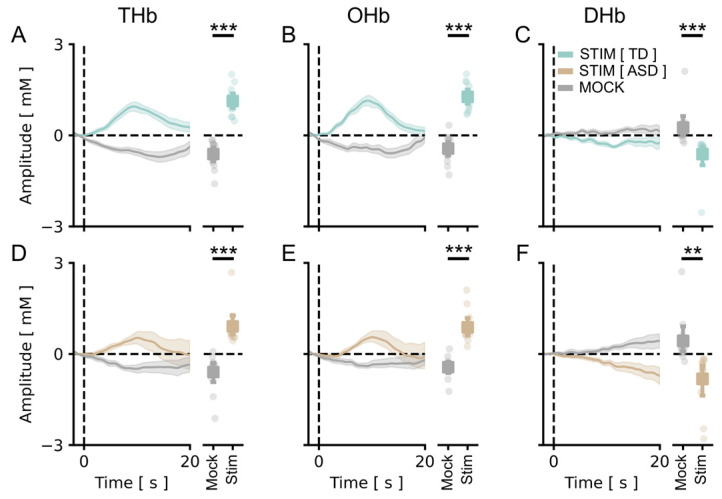
vHDR was reliably detected in TD and fASD participants: best channel. In all panels, values in the *y*-axis are multiplied by 10^4^. (**A**–**C**) On the left, the average time course for THb (**A**), OHb (**B**), and DHb (**C**) in response to the radial stimulus (Stim, green line) or the mock stimulus (Mock, grey line) are shown for TD participants. Plots on the right depict the average peak response to Stim vs. Mock across all of the subjects. The stimulus-driven signal was significantly higher with respect to the mock condition for all the metrics (*t*-test, *p* < 0.001 for all comparisons). (**D**–**F**) The same plots as above, for fASD subjects. The stimulus-evoked vHDR was also significantly higher with respect to the mock condition in this case (*t*-test, *p* < 0.001 for THb and OHb, *p* < 0.01 for DHb). The orange lines are the Hb responses to the stimulus in fASD plots. The data are expressed as mean ± sem. ** *p* < 0.01, *** *p* < 0.001.

**Figure 3 brainsci-13-00951-f003:**
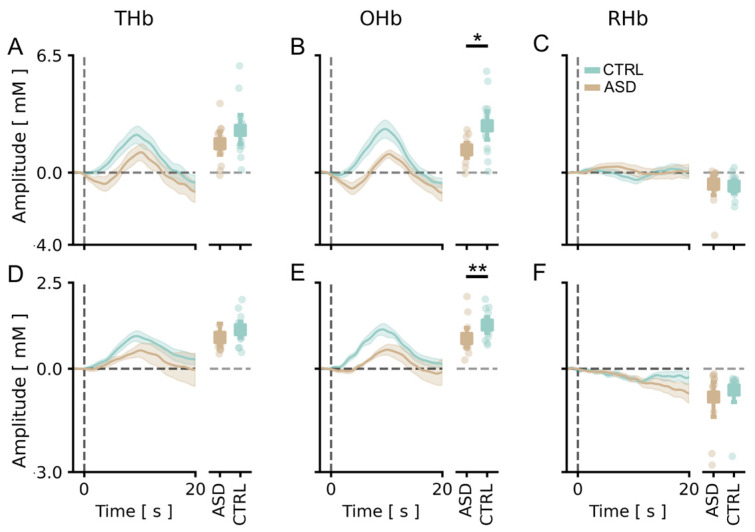
Comparison of vHDR between TD and fASD participants. In panels (**A**–**C**), values in the *y*-axis are multiplied by 10^5^; in panels (**D**–**F**), values in the *y*-axis are multiplied by 10^4^. (**A**–**C**) Average peak responses across all channels of THb (**A**), OHb (**B**) and DHb (**C**) in TD (green lines) and fASD participants (orange lines). An amplitude analysis revealed that the OHb peak was significantly lower in the fASD population than in controls (*t*-test, *p* < 0.05). (**D**–**F**) Peak response of THb (**D**), OHb (**E**) and DHb (**F**) in the best channel for TD (green lines) and fASD subjects (orange lines). The OHb peak was significantly lower in the fASD population than in the controls (*t*-test, *p* < 0.01). Data are expressed as mean ± sem. * *p* < 0.05, ** *p* < 0.01.

**Figure 4 brainsci-13-00951-f004:**
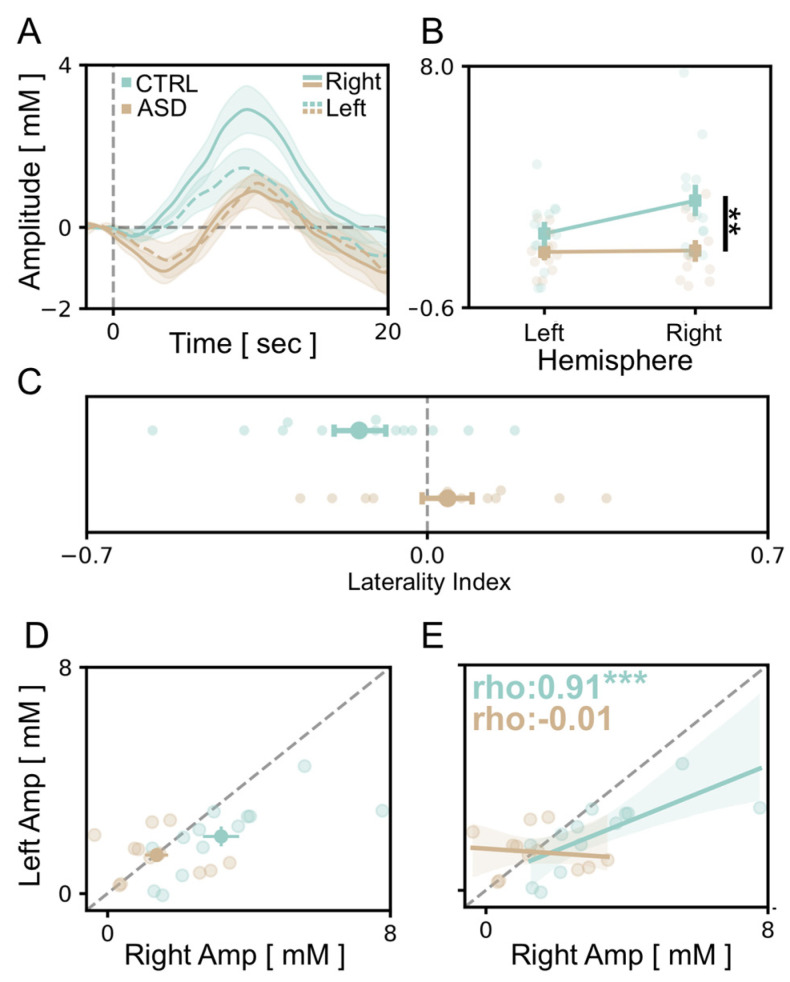
Atypical lateralization in visual processing in fASD children. (**A**) Average time course of OHb response in the left (dashed lines) and the right (solid lines) hemispheres for TD (green) and fASD subjects (orange line). The *y*-axis values are multiplied by 10^5^. (**B**) A comparison between the amplitude of the OHb signal in the left and the right hemisphere showed a significant reduction of right responses in the fASD group (a Two-Way mixed model ANOVA, *p* < 0.01). The *y*-axis values are multiplied by 10^5^. (**C**) The laterality index, indicating a lower rightward asymmetry of visual processing in fASD subjects compared to TD children (*t*-test, *p* < 0.05). (**D**) A scatterplot illustrating the distribution of average OHb amplitudes recorded in the right and the left hemisphere in TD and fASD subjects, showing a pronounced rightward bias only in TD children. (**E**) A linear regression analysis shows a significant correlation in the amplitude of average OHb responses evoked in the right and the left hemispheres for TD participants (Spearman correlation, *p* < 0.001). This correlation is absent in ASD patients (*p* = 0.965). The ρ (rho) index indicates the Spearman correlation value. In panels (**D**,**E**), the *y*-axis and *x*-axis values are multiplied by 10^5^. Levels of statistical significance are shown as ** *p* < 0.01; *** *p* < 0.001.

**Figure 5 brainsci-13-00951-f005:**
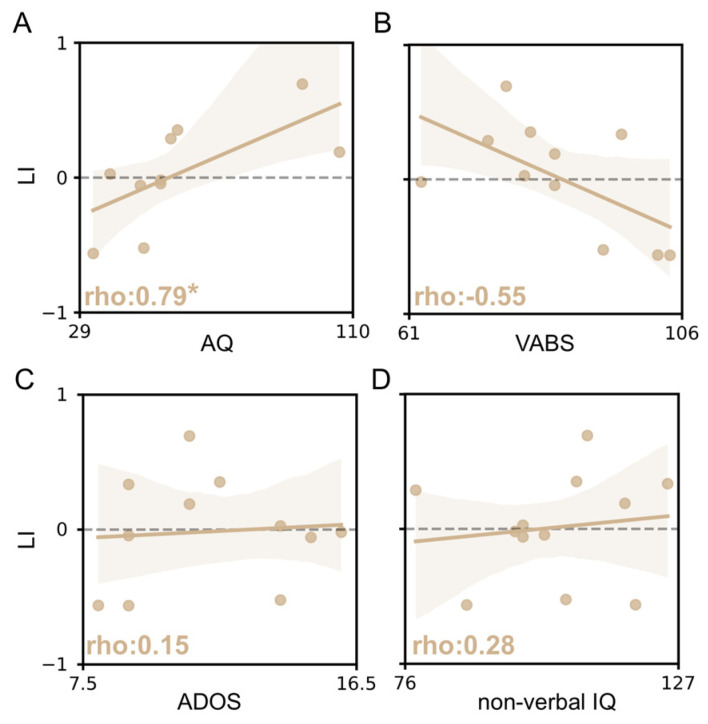
Correlation between the laterality index and clinical scores in fASD children. The ρ (rho) index in each plot indicates the Spearman correlation value. The correlation between Laterality Index (LI) and AQ score (**A**), VABS score (**B**), ADOS score (**C**) and non-verbal IQ score (**D**). A positive correlation between variables was detected for the AQ score (Spearman correlation, * *p* < 0.05).

**Table 1 brainsci-13-00951-t001:** Demographic data of the control group. Age (years), gender, head circumference (cm), cap size (cm), and the animated cartoon used for visual stimulation are listed for each participant (for movies, the production company, release date, and episode title are indicated as well).

ID	Age	Gender	Head	Cap Size	Cartoon Selected
A1	4	F	49	52	Frozen Walt Disney Pictures, 2013
A2	5	F	50.5	52	The Little Mermaid Walt Disney Pictures, 1989
A3	6	F	53	54	The Chipmunks, S7E04 “Bye, George” MWS and DIC Entertainment
A4	5	F	50.5	52	The Lion King Walt Disney Pictures,1994
A5	4	F	50.5	52	The Chipmunks, S7E04 “Bye, George” MWS and DIC Entertainment
A6	6	F	53	54	SpongeBob, S12E01 “FarmerBob” Nickelodeon Animation
A7	3	F	51	52	The Little Mermaid Walt Disney Pictures, 1989
A8	4	F	49.5	52	Frozen Walt Disney Pictures, 2013
A9	6	F	53	54	Inside out Disney-Pixar, 2015
A10	6	F	53	54	The Little Mermaid Walt Disney Pictures, 1989
A11	4	F	41.5	54	Frozen Walt Disney Pictures, 2013
A12	6	F	50.5	54	Frozen Walt Disney Pictures, 2013
A13	3	F	50	52	The Aristocats Walt Disney Productions, 1970

**Table 2 brainsci-13-00951-t002:** Demographic data of the fASD group. Age (years), gender, head circumference (head, cm), cap size (cm), and the animated cartoon used for visual stimulation are listed for each participant (for movies, the production company, release date, and episode title are indicated as well). Moreover, clinical variables collected, such as the ADOS total score and comparative score, the total AQ score, the non-verbal IQ score, and the VABS total score are listed for each subject. For non-verbal IQ scores, the psychometric scale which used for the evaluation is specified: the total score was used for the Leiter III scale; the performance score was used for the WPPSI III scale; the Perceptual Reasoning Index was used for the WISC IV scale; the mean score between “Foundations of learning” and “Eye and hand coordination” areas was used for GMDS III developmental assessment.

ID	Age	Gender	Head	Cap Size	Cartoon	ADOS_TOT	ADOS_comp	AQ_tot	nv_IQ	Cognitive Scale	VABS_tot
B1	4	F	51	52	Frozen Walt Disney Pictures, 2013	16	7	53	96	GMDS III	63
B2	5	F	50.5	52	Frozen Walt Disney Pictures, 2013	n.a.	n.a.	56	78	Leiter III	74
B3	6	F	53	54	Bing, S1E06 “Smoothie” Tandem Films e Digitales Studios, 2014	14	5	38	98	WPPSI III	80
B4	4	F	51	52	Frozen Walt Disney Pictures, 2013	9	5	n.a.	125	Leiter III	96
B5	3	F	48.5	52	Bing, S1E06 “Smoothie” Tandem Films e Digitales Studios, 2014	14	5	48	106	WPPSI III	93
B6	6	F	55	56	Uncle Grandpa, S3E04 “Uncle Easter” Cartoon Network Studios, 2016	12	7	58	108	WISC IV	81
B7	5	F	50	52	The Chipmunks, S7E04 “Bye, George” MWS and DIC Entertainment	9	6	96	87	GMDS III	87
B8	6	F	52	52	Frozen Walt Disney Pictures, 2013	11	6	106	117	WPPSI III	85
B9	5	F	51.5	52	Bing, S1E06 “Smoothie” Tandem Films e Digitales Studios, 2014	9	5	53	102	WPPSI III	85
B10	4	F	52	52	Frozen Walt Disney Pictures, 2013	15	5	47	98	GMDS III	n.a.
B11	3	F	53	52	Curious George S1E03 “Zeros to Donuts”, Universal Animation Studios, 2006	8	4	33	119	WPPSI III	102
B12	3	F	48.5	52	Masha and the Bear S1E26 “Home” Improvement, Animaccord Animation Studio	11	5	95	110	Leiter III	77

## Data Availability

The datasets generated during the current study are available on Zenodo: https://zenodo.org/record/7558643 (accessed on 22 January 2023).

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
