# Peer review of "A Potential Biomarker of Brain Activity in Autism Spectrum Disorders: A Pilot fNIRS Study in Female Preschoolers"

_brainsci, 2023, doi:10.3390/brainsci13060951_

Round 1
Reviewer 1 Report
Understanding brain circuitry in ASD is important to understanding the pathophysiology and to design better diagnostics and therapeutics. The authors use fNIRS in a small number of preschool girls, ASD versus TD, while watching a cartoon in order to determine differences in visual circuitry. Decreased amplitudes were noted in ASD, as expected, although the scatter plots show substantial overlap between the two groups, complicating the use of this methodology as a diagnostic test for ASD.
A major strength of this pilot study is the homogeneity of their participants, but of course that is a weakness in general application of the results. Although IQ was > 70, and TD participant IQ scores are not given, likely there are major differences in intelligence between the groups, and that might be more important than the presence or absence of autism in regards to the decreased amplitudes noted. A (brief) acknowledgement of this limitation in the Discussion is warranted.
The laterality data is very interesting, yet my interpretation is quite different. We agree that the scatter plots clearly show right-orientation in TD, although the laterality is minimal in some. However, the authors claim that ASD is non-lateralized. My reading of their data is that some are right-lateralized (perhaps even more than TD?), some are minimally lateralized, and some are left lateralized. I am interested in any differences between the right- and left-lateralized ASD participants, but likely more data is needed to answer that.
I hope that this group will follow this pilot study with a much larger study of both genders, ASD, ID, and TD, and to query for differences that mark the outliers.
Reviewer 2 Report
This research is excellent and intriguing, with a high degree of novelty. The only limitation of this study was the small sample size. However, the authors have explained the study's limitations. Therefore, I believe this article should be published and expanded in the future.
